# *Anaplasma* and *Ehrlichia* Species in *Ixodidae* Ticks Collected from Two Regions of Bulgaria

**DOI:** 10.3390/microorganisms11030594

**Published:** 2023-02-26

**Authors:** Iskren Stanilov, Alexander Blazhev, Lyuba Miteva

**Affiliations:** 1Department of Molecular Biology, Immunology and Medical Genetics, Faculty of Medicine, Trakia University, 6000 Stara Zagora, Bulgaria; 2Department of Biology, Medical University-Pleven, 5800 Pleven, Bulgaria

**Keywords:** *E. canis*, *A. phagocytophilum*, *Anaplasmataceae*, species-specific PCR

## Abstract

The aim of the study was to determine prevalence of *Anaplasmataceae*-infected ticks in the Black Sea Coast and the Pleven regions of Bulgaria. A total of 350 ticks from different tick species were collected. Two hundred fifty-five ticks were removed from dogs in the Black Sea Coast region, and 95 *Ixodes ricinus* ticks were collected by flagging vegetation with a white flannel cloth in two areas in the region of Pleven. After the DNA isolation of the ticks, a genus-specific polymerase chain reaction (PCR) was performed to identify *Anaplasmataceae.* Second PCRs were performed with species-specific primers to identify *Ehrlichia canis* (*E. canis*) and *Anaplasma phagocytophilum* (*A. phagocytophilum*). The results showed that 26.9% of the *Ixodes ricinus* ticks were infected with *Anaplasmataceae* in the Black Sea Coast region and 36.8% in the Pleven region. The infection with *E. canis* was detected in 35.7% and *A. phagocytophilum* in 25.0% of positive ticks from the Black Sea Coast region. In the Pleven region, 22.9% of ticks were positive for *E. canis*, while 42.9% were positive for *A. phagocytophilum*. The molecular identification of *E. canis* in ticks collected from Bulgaria was performed for the first time. In conclusion, the present study revealed a higher prevalence of ticks infected with *Anaplasmataceae*, particularly *A. phagocytophilum*, in the Pleven region than in the Black Sea Coast region.

## 1. Introduction

Ticks represent the second vector group after mosquitoes, consisting of more than 900 tick species worldwide that transmit a broad range of pathogens to domestic animals and humans [1]. The expansion of their geographical area and the increasing incidence of tick-borne diseases over the last 20 years have significantly influenced domestic animal and human health. The most important tick species responsible for zoonotic disease transmission and spreading belong to the *Ixodidae* family [2].

Their successful survival depends on optimal temperature and humidity in their environment and the availability of appropriate hosts [3,4]. During blood feeding, ticks can transmit viruses, bacteria, and protozoan parasites to their hosts, mainly birds and mammals, including humans. Ticks and tick-transmitted pathogens have co-evolved with various wild animal hosts. As a result, many of these wild animals have become reservoir hosts and successfully live in a state of equilibrium with tick-borne pathogens [5]. That situation facilitates the spreading of tick-borne diseases in pets and humans. Moreover, tick species are recently present in more areas than ever, including urban green regions (city parks and gardens) and peri-urban forest areas, facilitating contact between humans and pets and infected questing ticks [6,7]. Among ixodid ticks, *Ixodes ricinus* is the most known species in terms of spreading and significant transmission in Europe and Bulgaria [6,8]. Moreover, transstadial transmission of pathogens in the *Ixodes ricinus* developmental stages is important for their epidemiological significance.

Among the bacterial pathogens transmitted by ixodid ticks, *Borrelia burgdorferi* s.l., *Anaplasma*, and *Ehrlichia* spp. significantly influence human health. The family *Anaplasmtaceae* comprises obligate intracellular Gram-negative bacteria and includes agents of *Ehrlichia* and *Anaplasma* species. *Anaplasma phagocytophilum* causes canine and human granulocytic anaplasmosis. *Ehrlichia canis* cause canine monocytic ehrlichiosis, and some cases of human infection were described [9]. Moreover, during the last decade, incidence of tick-borne diseases in Europe has increased [6]. For the last two decades, Bulgarian studies on tick-borne pathogens, particularly *Anaplasmataceae*, have been scarce. Little is known about the prevalence of *Ehrlichia* and *Anaplasma* spp. in ticks from different regions of Bulgaria [10,11].

In the present study, two distanced regions of Bulgaria were explored: the Black Sea Coast of Eastern Bulgaria and the Pleven region of North Bulgaria. Both regions provide the optimal environment for tick development in temperature, humidity, and potential hosts. In addition, many people frequently visit these areas. All of these epidemiological characteristics of the above-described areas make it reasonable to explore tick density and tick-borne pathogen occurrence within these regions. To date, no study has been carried out on the presence of *Anaplasmataceae* pathogens in ticks in these places in Bulgaria. Therefore, this study aims to detect the presence of *Anaplasmataceae* bacterial DNA in ticks and to establish the prevalence of *Anaplasma phagocytophilum* (*A. phagocytophilum*) and *Ehrlichia canis* (*E. canis*) in areas of Byala, St. Vlas, and the Pleven region of Bulgaria.

## 2. Materials and Methods

### 2.1. Sample Collection

The ticks (n = 350) were collected from two Bulgarian sites: Byala and Sveti Vlas on the Black Sea Coast and the Pleven region (Kaylaka park and Kartozhabene) between 2021 and 2022 (Figure 1). Geographic mapping was conducted on Quantum Geographic Information Systems (QGIS version 3.18.0, QGIS Development Team, GNU General Public License, Essen, Germany) with the World Geodetic System 1984 (WGS 84) being a standard of coordinate referencing.

The towns of Byala and St. Vlas are located on the Black Sea Coast of Eastern Bulgaria. This region covers the mountainous part of the Black Sea Coast, where the Balkan Mountains (Stara Planina) reach the Black Sea. It is a forested area favourable for *Ixodes* species. These towns are seaside resorts that are visited by many people, especially during the summer season. A total of 255 ticks were removed from stray dogs in an area of Byala town (n = 27) and domestic dogs from private veterinary clinics in St. Vlas town (n = 15). Collected ticks were maintained alive in individual 1.5 mL Eppendorf tubes and transported to the Department “Molecular Biology, Immunology and Medical Genetics” laboratory at the Medical Faculty, Trakia University, for species identification and testing for pathogens. In the laboratory, all ticks underwent taxonomic identification, developmental staging, and sexing, according to Georgieva and Gencheva [12] and Estrada-Peña et al. [13]; the identification was performed using a stereomicroscope (Olympus SZ4045, Olympus American Inc., Melville, NY, USA). Using the morphology of the mouthparts (length of the palps to the basis capituli and the shape of the basis capituli); the presence of festoons and specific patterning (ornate) on the dorsal shield; the presence of eyes; the shape of the anal groove; and the shape of the coxae, we identified 104 ticks belonging to the genus *Ixodes* and 151 ticks from the genus *Rhipicephalus.* The collected *Ixodes* ticks included 46 (44.2%) fed, 32 (30.8%) semi-fed and 26 (25%) unfed ticks or nymphs.

The second studied region is close to Pleven in Northern Bulgaria and includes Kaylaka Park and the village of Kartozhabene. Kaylaka Park is located near the town of Pleven and is a frequently visited place by many people. The village of Kartozhabene is around 10 km from the town of Pleven. Both spots in the Pleven region are in the karst canyons of rivers, with abundant and diverse flora and fauna, and are favourite places for outdoor activities for many visitors. A total of 95 *Ixodes* ticks were collected from the vegetation by flagging in urbanized and wild areas in the region of Pleven. A detailed description of sampling from the Pleven region was previously described [14].

### 2.2. DNA Extraction and Polymerase Chain Reaction (PCR) Identification of Tick-Borne Bacteria

Total DNA was isolated from all collected *Ixodes* ticks (n = 199), and from 44 randomly selected ticks from the genus *Rhipicephalus*, using the animal tissue genomic DNA purification mini prep kit (Gennaxxon bioscience, Ulm, Germany) or the NucleoSpin Tissue, Mini kit for DNA from cells and tissue (Macherey-Nagel KG; Düren, Germany), following the manufacturer’s instructions. The DNA extraction kits have comparable performance. The extracted DNA was stored at −70 °C until PCR analysis could be conducted.

Firstly, the gradient PCR was performed in an AERIS PCR system (Esco, Singapore) to optimize the annealing temperatures in any PCR. Then, each DNA sample was tested first for the presence of the 16S rRNA consensus sequence specific for all bacterial species of the family *Anaplasmataceae*, with a conventional PCR, using the primers presented in Table 1 and previously described by Inokuma et al. [15]. For the PCR run, we used 3 µL template DNA and the reaction mixture (20 µL) contained 2 µL 10XPCR buffer, 1.5 mM MgCl_2_, 0.2 mM each of dNTPs, 0.25 μM each of primers, and 1U Taq DNA polymerase. The amplification was performed as follows: 96 °C for 3 min, 35 cycles of 95 °C for 30 s, 55 °C for 30 s and 40 s at 72 °C, and a final extension at 72 °C for 7 min.

Subsequently, positive samples were tested with a second PCR using a species-specific primer set for *Ehrlichia canis* (*E. canis*) or *Anaplasma phagocytophilum* (*A. phagocytophilum*) and the annealing temperature presented in Table 1 [16,17]. The 5mM MgCl_2_ was used in a species-specific PCR for *E. canis.* The rest condition of the reaction mix was the same as the first PCR. The used PCR components were manufactured by Thermo Fisher Scientific, Vilnius, Lithuania, and the primers by Metabion GMBH, Planegg, Germany.

All PCR products were analysed on a 1.5% agarose gel electrophoresis stained with ethidium bromide. A DNA Ladder (by 100 bp) was applied for evaluation of the obtained product size. The results of the PCR amplification were viewed under UV light and were archived using EasyWin32 software (Herolab; Wiesloch, Germany). DNA extraction, fragment amplification, and agarose gel electrophoresis were performed in separate rooms.

### 2.3. Statistical Analysis

The frequency of data is reported as counts and as the percentage of the total number of ticks or the percentage of the *Anaplasmataceae*-positive *Ixodes ricinus* ticks in the regions that were studied. Differences in tick infection prevalence between sexes and localities were analysed using a nonparametric Chi-square test and 95% confidential interval (95% CI). The differences were considered significant with *p* values less than 0.05.

## 3. Results

The ticks collected from dogs in regions of Byala and St. Vlas were mainly from the genus *Rhipicephalus* (n = 151) and *Ixodes* (n = 104). The collected *Ixodes* ticks included 54 (51.9%) female ticks, 33 (31.7%) male ticks, and 17 (16.4%) nymphs. The ticks collected from the Pleven region (n = 95) were from the genus *Ixodes*, including 46 (48.4%) females, 44 (46.3%) males, and 5 (5.3%) nymphs. All *Ixodes* ticks (n = 199) were classified as *Ixodes ricinus* and were subjected to further analysis to detect the prevalence of bacteria of the *Anaplasmataceae* family.

We did not detect any infected *Rhipicephalus sanguineus* ticks with bacteria in the *Anaplasmataceae* family among the randomly selected 44 ticks. As a result, these tick species were excluded from further analysis.

The overall prevalence of bacteria in the *Anaplasmataceae* family in *Ixodes ricinus* ticks collected from the Black Sea Coast and Pleven regions are presented in Figure 2. Although there was a slightly higher rate of infected *Ixodes* ticks in the Pleven region (36.8%, 35 of 95) in comparison to the Black Sea Coast region (26.9%, 28 of 104), the statistical significance was not reached (*χ*^2^ = 2.258; *df* = 1; *p* = 0.133). The overall prevalence of ticks infected with bacteria in the *Anaplasmataceae* family is 31.7% (63 of 199).

As is shown in Table 2, the highest rate of infection with the *Anaplasmataceae* pathogens was detected amongst females, followed by male ticks in both of the studied regions. None of the tested nymphs was infected with species of *Anaplasmataceae*. In the Black Sea Coast region, 19 of 54 female ticks (35.2% of females) and 9 of 33 male ticks (27.3% of males) were infected (*χ^2^* = 0.588; *df* = 1; *p* = 0.443). In the Pleven region, there was a 14-fold increase of infected female ticks compared to male ticks (OR = 14.6; 95% CI: 4.814 ÷ 44.434; *p* < 0.0001). Here, 30 of the 46 female ticks (65.2% of females) were infected with the *Anaplasmataceae* species, which was significantly higher than the infection rate seen in male ticks from the same area (5 of 44 males, 11.4% of males with *χ^2^* = 27.444; *df* = 1; *p* < 0.001).

Since the ticks collected from the Black Sea Coast region were removed from dogs during their feeding, we compared the prevalence of infected ticks with bacteria in the *Anaplasmataceae* family across the subgroups, taking into account their feeding status. Among the female infected ticks, 13 of 19 (68.4%) were fed, 4 (21.1%) were semi-fed, and 2 (10.5%) were unfed ticks. Similarly, among the male infected ticks 4 of 9 (44.4%) were fed, 2 (22.2%) were semi-fed, and 3 (33.3%) were unfed ticks (*χ^2^* = 2.36; *df* = 2; *p* = 0.307). Altogether, the highest infective rate was detected among fed ticks (60.7%; 17 of 28), followed by semi-fed (21.4%; 6 of 28) and unfed (17.9%; 5 of 28) ticks.

Ticks that tested positive in the test for the detection of pathogens from the *Anaplasmataceae* family were further analysed with a species-specific PCR assay for *E. canis* and *A. phagocytophilum*. The observed results are presented in Table 3.

In the Black Sea Coast region, among the *Anaplasmataceae*-positive ticks, 35.7% (10 of 28) were infected with *E. canis* and 25.0% (7 of 28) with *A. phagocytophilum*. Co-infection was observed in 10.7% (3 of 28) and 39.3% (11 of 28) were negative for both of the pathogens studied.

In the Pleven region, *E. canis* was identified in 8 ticks (22.9% of *Anaplasmataceae*-positive ticks and 8.6% for all tested ticks), and 15 ticks (42.9% of *Anaplasmataceae*-positive ticks and 15.8% of for all tested ticks) were positive for *A. phagocytophilum*. Co-infection was observed in 17.1% (6 of 35), and 34.3% (12 of 35) were negative for both of the pathogens studied.

The infection rate of *E. canis* between the two studied regions was similar when considering only *Ananplasmataceae*-positive ticks (35.7%; 10 of 28 vs. 22.9%; 8 of 35 with *χ^2^* = 1.260; *df* = 1; *p* = 0.262), as well as when considering all the tested ticks (9.6%; 10 of 104 vs. 8.4%; 8 of 95; *χ^2^* = 0.086; *df* = 1; *p* = 0.769).

A difference in the infection rate of *A. phagocytophilum* between the two studied regions was observed. The infection rate amongst *Anaplasmataceae*-positive ticks was seen to be higher in the Pleven region than in the Black Sea region (42.9%; 15 of 35 vs. 25%; 7 of 28; *χ^2^* = 2.183; *df* = 1; *p* = 0.140). The difference reached a statistical significance when considering all of the tested ticks within this study (15.8%; 15 of 95 vs. 6.7%; 7 of 104; *χ^2^* = 4.143; *df* = 1; *p* = 0.042). In the Black Sea region, the infection rate of *E. canis* was more common than infection with *A. phagocytophilum*, contrary to the Pleven region (Figure 3).

The co-infection rate was similar in both of the areas studied. Co-infection with *E. canis* and *A. phagocytophilum* was detected in 3 of 28 (10.7%) ticks from the Black Sea Coast region and 6 of 35 (17.1%) ticks from the Pleven region, with a *χ^2^* = 0.525; *df* = 1; *p* = 0.469. A total of 14.3% of ticks were infected with both of the pathological species from the *Anaplasmataceae* family.

## 4. Discussion

Although accurate prediction by epidemiological data is not possible, they can reveal which areas are more prone to tick-borne diseases and could potentially be used for minimizing the risk of transmission of these diseases. Molecular tools such as PCR identification can provide a better understanding of the epidemiology of target pathogens.

The main investigated tick species was *Ixodes ricinus*, the most common and widely distributed tick in Europe and Bulgaria [18,19,20]. Furthermore, the prevalence of infection with *A. phagocytophilum* and *B. burgdorferi* in rodents within Bulgaria clarifies their role as competent reservoirs for these pathogens in nature [21].

In the present study, PCR techniques were used to detect *Ehrlichia canis* and *Anaplasma phagocytophilum*, which represent an emerging threat to public health due to their zoonotic nature. We investigated the rate of bacterial infection of *Ixodes* ticks, which were collected from two distant regions in Bulgaria, since both areas are frequently visited by locals and tourists. Byala and St. Vlas are seaside resorts in Central-Eastern Bulgaria, located on the Black Sea Coast, and lie in a semi-mountainous region in the eastern region of Stara Planina. Kaylaka Park is located on the periphery of the regional town of Pleven and is a local place for walks and leisure activities, with a large number of daily visitors. The village of Kartozhabene is located twelve kilometres southwest of Pleven. Our results showed the difference in infected ticks between the two studied regions. The frequencies of the ticks infected with bacteria belonging to the family *Anaplasmataceae* are higher in the Pleven region (36.8%) than in the region of the Black Sea Coast (26.9%), though they do not display statistical significance. Also, we detected DNA from *Anaplasmataceae* pathogens in *Ixodes*, but not in *Rhipicephalus*, ticks which confirms the results from other authors that *Ixodes* are the predominant vector for these bacteria in Europe and, therefore, in Bulgaria [6,10].

Two genera from the family *Anaplasmataceae—Ehrlichia* and *Anaplasma*—comprise tick-borne intra-cellular, non-motile, Gram-negative bacteria of medical and veterinary concern. They parasitize in human and canine neutrophils (*Anaplasma* spp.) or monocytes and macrophages (*Ehrlichia* spp.). *E. canis* is a causative agent of canine monocytic ehrlichiosis (CME) and is transmitted mainly by two genera of ticks, *Ixodes* and *Rhipicephalus* [3,22]. It was previously thought that this bacterium only infected dogs, until the publication of Perez et al. [9]. Other recent reports confirm that the etiological agent of CME *E. canis* can also be a human pathogen [23,24]. The molecular detection of *E. canis* in *Ixodes* ticks in the two investigated regions in Bulgaria was similar and varied between 8.4% (Pleven) and 9.6% (Black Sea Coast) from all of the tested ticks. To our knowledge, no other study has examined the presence of *E. canis* DNA in ticks from Bulgaria. The published data refers to the study of antibodies against *E. canis* in dog serum, with total seropositivity of dogs from southern Bulgaria at approximately 30%, with a variation between 15% and 50% for the different regions studied [25,26]. The distribution of *E. canis* in European countries varies greatly, which has been seen in the published work [27].

Among six species of the genus *Anaplasma*, only *A. phagocytophilum* causes a febrile disease in humans and other mammals [28]. The human infection with *A. phagocytophilum* has been referred to as human granulocytic anaplasmosis (HGA). *A. phagocytophilum* is mainly transmitted by *I. ricinus*, and our results confirm that none of the species belonging to *R. sanguineus* was positive for this pathogen. Even in ticks removed from a single animal, Anaplasma infection was not detected in co-feeding *Ixodes* and *Ripicephalus* ticks. The molecular detection of *A. phagocytophilum* was performed by detecting the *ankA* gene, which encodes a protein characterized by repeated ankyrin motifs and is involved in the molecular mechanism of host adaptation selection [29].

The prevalence of infected ticks with *A. phagocytophilum* in the Pleven region was higher than that found in the Black Sea Coast region, reaching the statistical significance when taking into account all of the tested ticks (*χ^2^* = 4.143; *p* = 0.042). The co-infection was also higher for the Pleven region in comparison to the Black Sea Coast region (17.14% vs. 10.7%). These results are supported by the data on the spread of *A. phagocytophilum* in rodents (average 8.8% for rodents trapped in various regions of Bulgaria), which are the main natural reservoir [21]. Another recent study for the molecular detection of *A. phagocytophilum* in *Ixodes* ticks, collected from Strandja Nature Park in south-eastern Bulgaria [11], reported a significantly higher prevalence of that pathogen in *Ixodes* spp. (38.9%). In our study, a total of 34.9% (22 of 63) of *Ixodes* ticks were infected with *A. phagocytophilum*, which is similar to the rate of 33.9% reported previously among adult ticks collected near Sofia, Bulgaria [10].

All of this data collectively demonstrates the abundant distribution of *A. phagocytophilum* in ticks and animals in Bulgaria, but, surprisingly, Anaplasmosis is only occasionally detected in patients. The lack of correlation between the prevalence of *A. phagocytophilum* and the number of diagnosed patients can be explained by the unspecific clinical manifestation of Anaplasmosis, incomplete diagnosis, or a high rate of asymptomatic infections. We can suspect that Anaplasmosis is much more common in Bulgaria than the official data presented for that disease. Most European cases of Anaplasmosis are presented as a mild infection, and the suspicion of being under-diagnosed or under-reported remains. Data for *A. phagocytophilum*-seropositive humans in Europe is on average ~8.3%, reaching up to 31% [30]. Some serological studies reported the presence of both HGA and Lyme borreliosis antibodies, suggesting co-infection transmitted from a tick’s bite, since the vector is the same [30]. Accurate molecular tests should be recommended for patients before the treatment to avoid antibiotic overuse, and for blood donors because a blood transfusion can also transmit this pathogen.

Finally, it should be mentioned when using the method of molecular testing, many of the ticks positive for 16S rRNA of *Anaplasmataceae* are negative for both *E. canis* and *A. phagocytophilum*. This can be due to the presence of other pathogens in that family, and some of them can have a zoonotic potential and can be a future threat to human health. All of this requires further investigation to determine the epidemiological significance of tick-borne diseases.

## 5. Conclusions

In conclusion, our study presents epidemiological data for the rate of infection with pathogens belonging to the family of *Anaplasmataceae* in *Ixodes ricinus* ticks collected from two regions of Bulgaria: the Pleven and Black Sea Coast regions. The detection was performed by molecular identification of the pathogens’ DNA sequences. Generally, the Pleven region presents a higher frequency of infected ticks with studied pathogens. Although the fed or semi-fed ticks may be newly infected, since it is not known whether the pathogens’ DNA was in a tick or from the animal’s blood, all ticks collected from the region of Pleven were collected from the vegetation by flagging and were unfed. Therefore, we may accept the Pleven region as a site with a higher prevalence of ticks infected naturally with *Anaplasmataceae* than the Black Sea Coast region. In addition, the distribution of *Ehrlichia canis* in ticks from the two regions did not reach a significant difference, while the prevalence of *Anaplasma phagocytophilum* in the Pleven region was significantly higher than that found in the Black Sea coast region.

## Figures and Tables

**Figure 1 microorganisms-11-00594-f001:**
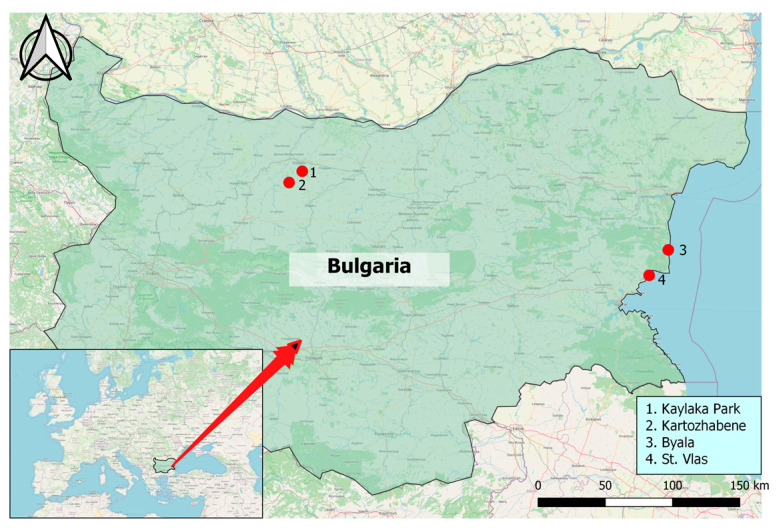
Location of areas for tick collection. Pleven region in Northern Bulgaria including (1) Kaylaka park and (2) village Kartozhabene, as well as the Black Sea Coast region of Eastern Bulgaria, including the towns of (3) Byala and (4) St. Vlas.

**Figure 2 microorganisms-11-00594-f002:**
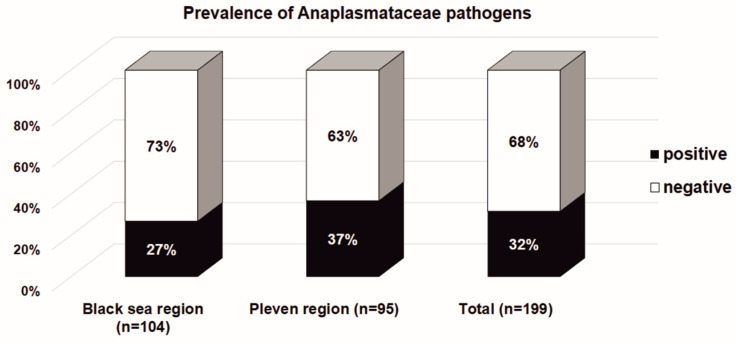
The *Ixodes ricinus* infectivity with bacteria in the *Anaplasmataceae* family in the Black Sea Coast and Pleven regions. The results are presented as a percentage of the total number of tested *Ixodes ricinus* ticks.

**Figure 3 microorganisms-11-00594-f003:**
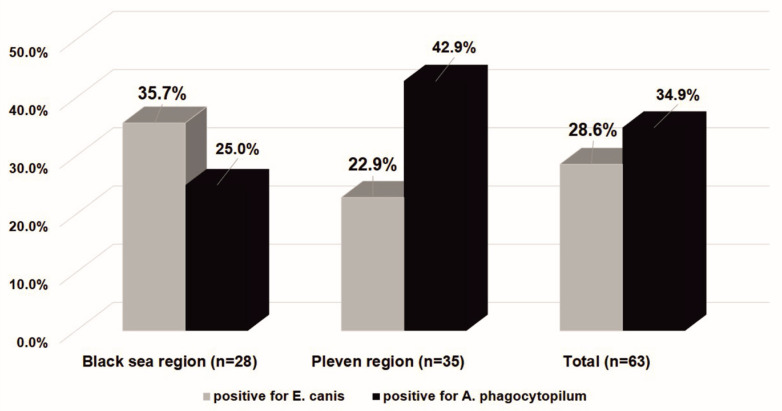
The *Ixodes ricinus* infectivity with bacteria *E. canis* and *A. phagocytophilum* in the Black Sea Coast and Pleven regions. The results are presented as a percentage of the number of ticks positive for bacteria in the *Anaplasmataceae* family.

**Table 1 microorganisms-11-00594-t001:** Primers and PCR conditions used for molecular identification of pathogens from family *Anaplasmataceae* and species identification of *E. canis* and *A. phagocytophilum*.

Primer Name	Target Gene	Target Sequence (5′-3′)	Amplicon Size	Annealing Temperature
Amplification of “consensus” sequences of family *Anaplasmataceae*				
EHR16SD	16S rRNA	GGTACCYACAGAAGAAGTCC	345 bp	55 °C
EHR16SR		TAGCACTCATCGTTTACAGC		
Amplification of species-specific sequences of *E. canis*				
Canis	16S rRNA	CAATTATTTATAGCCTCTGGCTATAGGA	409 bp	63 °C
GA1UR R		GAGTTTGCCGGGACTTCTTCT		
Amplification of species-specific sequences of *A. phagocytophilum*				
LA1 F	AnkA	GAGAGATGCTTATGGTAAGAC	444 bp	44.2 °C
LA6 R		CGTTCAGCCATCATTGTGAC		

**Table 2 microorganisms-11-00594-t002:** The infective rate of *Ixodes ricinus* ticks with bacteria in the *Anaplasmataceae* family, according to their stage and sex in studied regions.

Region	Stage and Sex of *I. ricinus*			PCR-Positive Ticks for Bacteria of the *Anaplasmataceae* Family			
	Female, n (%)	Male, n (%)	Nymph, n (%)	Female, n (%)	Male, n (%)	Nymph, n (%)	Total, n (%)
Byala (n = 92)	46 (50.0)	30 (32.6)	16 (17.4)	17 (18.5)	8 (8.7)	0	25 (27.2)
St. Vlas (n = 12)	8 (66.7)	3 (25.0)	1 (8.3)	2 (16.7)	1 (8.3)	0	3 (25.0)
Black Sea Coast-total (n = 104)	54 (51.9)	33 (31.7)	17 (16.4)	19 (18.3)	9 (8.7)	0	28 (26.9)
Kaylaka Park (n = 84)	42 (50.0)	37 (44.0)	5 (6.0)	27 (32.1)	5 (5.95)	0	32 (38.1)
Kartozhabene (n = 11)	4 (36.4)	7 (63.6)	0	3 (27.3)	0	0	3 (27.3)
Pleven-total (n = 95)	46 (48.4)	44 (46.3)	5 (5.3)	30 (31.6)	5 (5.3)	0	35 (36.8)

The presented percentages are calculated for the total number of the tested ticks.

**Table 3 microorganisms-11-00594-t003:** The prevalence of *E. canis* and *A. phagocytophilum* among the *Anaplasmataceae*-positive *Ixodes ricinus* ticks in studied regions.

Region	Ticks Positive for Bacteria of *Anaplasmataceae* Family	Ticks Positive for *E. canis*		Ticks Positive for *A. phagocytophilum*		Co-Infected Ticks	
	n	n	%	n	%	n	%
Byala (n = 92)	25	10	40.0	6	24.0	3	12.0
St. Vlas (n = 12)	3	-	-	1	33.3	-	0.0
Black Sea Coast-total (n = 104)	28	10	35.7	7	25.0	3	10.7
Kaylaka Park (n = 84)	32	8	25.0	14	43.8	6	18.8
Kartozhabene (n = 11)	3	-	-	1	33.3	-	0.0
Pleven-total (n = 95)	35	8	22.9	14	42.9	6	17.1

The presented percentages are calculated for the number of ticks positive for bacteria in the *Anaplasmataceae* family.

## Data Availability

Not applicable.

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
