# Peer review of "Anaplasma and Ehrlichia Species in Ixodidae Ticks Collected from Two Regions of Bulgaria"

_microorganisms, 2023, doi:10.3390/microorganisms11030594_

Round 1

Reviewer 1 Report

The paper has its strength being a first report, although generally tick sample size is small,

The different tick species collected should be clarified in the abstract

Line 8; The aim of the study was to “detect the prevalence”? Should be to “determine prevalence”

Line 11; 95 Ixodes ricinus ticks were collected by flagging.? Flagging what sites/where?

Line 14-15; The results showed that 26.9% of the Ixodes ricinus ticks were infected with Anaplasmataceae in the Black Sea Coast region

Author Response

ANSWER TO THE REVIEWERS

Dear Reviewers and Editor,

Thank you very much for your time and efforts for the valuable comments and remarks on our manuscript. We considered all of them very carefully and made all the recommended changes, which in our opinion significantly improved the quality of the article. All changes in the article are highlighted in yellow.

Here are the changes made in accordance with the recommendations, as well as the answers to the additional issues:

 REVIEWER 1

  1. The different tick species collected should be clarified in the abstract

 Authors: The different ticks collected was clarified in the abstract as follows:

“A total of 350 ticks from different tick species were collected.”

  1. Line 8; The aim of the study was to “detect the prevalence”? Should be to “determine prevalence”

Authors: The aim of the study was changed to “determine prevalence”.

  1. Line 11; Ixodes ricinus ticks were collected by flagging.? Flagging what sites/where?

Authors: The flagging site was clarified in the abstract as follows:

“Two hundred fifty-five ticks were removed from dogs in the Black Sea Coast region, and 95 Ixodes ricinus ticks were collected by flagging vegetation with a white flannel cloth from two areas in the region of Pleven.”

  1. Line 14-15; The results showed that 26.9% of the Ixodes ricinusticks were infected with Anaplasmataceae in the Black Sea Coast region

Authors: The sentence: “The results showed 26.9% infected Ixodes ricinus ticks with Anaplasmataceae in the Black Sea Coast region…” was changed to:

“The results showed that 26.9% of the Ixodes ricinus ticks were infected with Anaplasmataceae in the Black Sea Coast region….”

Reviewer 2 Report

The article is very interesting. It provides information on the occurrence of tick-borne pathogens in two regions of Bulgaria. The work is valuable and important for monitoring the environment for potential human and animal infections. This work is good, however, it needs corrections and rethinking a few issues.

Comments:

Introduction: you can add some more information about detected tick-borne pathogens;

Materials and methods:

1. Were the ticks identified only morphologically (appropriate keys)? There may be similar species. It is worth performing additional molecular identification.

2.  Were the collected ticks fed/semi fed? It matters for the results.

Results and conclusions:

1. The agarose gel with results is not necessary. This is rarely practiced today.

2. It is not clear whether the results refer to ticks fed/semi fed/unfed or collected from were vegetation.

3. The results for fed ticks may be false positive - it is not known whether the pathogen DNA was in the tick or from the animal's blood.

This should be included in the conclusion.

Hence, I would rethink the paper title. In my opinion it should be changed.

Author Response

ANSWER TO THE REVIEWERS

Dear Reviewers and Editor,

Thank you very much for your time and efforts for the valuable comments and remarks on our manuscript. We considered all of them very carefully and made all the recommended changes, which in our opinion significantly improved the quality of the article. All changes in the article are highlighted in yellow.

Here are the changes made in accordance with the recommendations, as well as the answers to the additional issues:

REVIEWER 2

Introduction: 

  1. You can add some more information about detected tick-borne pathogens;

Authors: page 2, line 2. Additional information about the studied pathogens was added in the Introduction as follows:

“The family Anaplasmtaceae comprises obligate intracellular Gram-negative bacteria and includes agents of as Ehrlichia, and Anaplasma species. Anaplasma phagocytophilum causes canine and human granulocytic anaplasmosis. Ehrlichia canis cause canine monocytic ehrlichiosis and some cases of human infection were described [9]. “

The numbering of the references was update.

 Materials and methods:

  1. Were the ticks identified only morphologically (appropriate keys)? There may be similar species. It is worth performing additional molecular identification.

Authors: The ticks were identified only morphologically according to the two qualifiers - first to Estrada-Peña, and second to Georgieva [ref 12 and 11]. In addition, two examiners independently performed species identification. We are agreeing with the reviewer’s suggestion for the molecular identification of ticks and will take into consideration for the further studies, but all Ixodes species have the same potential to be a vector for studied pathogens.

  1. Were the collected ticks fed/semi fed? It matters for the results.

Authors: The additional information about the feeding status of the collected ticks was added in 2. Materials and Methods section, 2.1. Sample collection, page 3, as follows:

“The collected Ixodes ticks include 46 (44.2%) fed, 32 (30.8%) semi-fed and 26 (25%) unfed ticks or nymphs.”

Results and conclusions:

  1. The agarose gel with results is not necessary. This is rarely practiced today.

Authors: The figure 2 was deleted. Respectively the figures’ numbers was update.

  1. It is not clear whether the results refer to ticks fed/semi fed/unfed or collected from were vegetation.

Authors: An additional comparison in infection rate with Anaplasmataceae pathogens between fed, semi-fed and unfed ticks was performed and the observed results were added in section Results, page 5, last paragraph, as follows:

“Since the ticks collected from the Black Sea coast region were removed from dogs during their feeding, we compared the prevalence of infected ticks with bacteria in the Anaplasmataceae family across the subgroups taking into account their feeding status. Among the female infected ticks, 13 of 19 (68.4%) were fed, 4 (21.1%) were semi-fed and 2 (10.5%) were unfed ticks. Similarly, among the male infected ticks 4 of 9 (44.4%) were fed, 2 (22.2%) were semi-fed and 3 (33.3%) were unfed ticks (χ2 = 2.36; df =2; p = 0.307). Altogether, the highest infective rate was detected among fed ticks (60.7%; 17 of 28) followed by semi-fed (21.4%; 6 of 28) and unfed (17.9%; 5 of 28) ticks.”

In addition, in the Discussion section was added information about Anaplasma infection among co-feeding ticks on page 8 as follows:

“Even in ticks removed from a single animal, Anaplasma infection was not detected in co-feeding Ixodes and Ripicephalus ticks.”

  1. The results for fed ticks may be false positive - it is not known whether the pathogen DNA was in the tick or from the animal's blood.

This should be included in the conclusion.

Authors: The following was included in the Conclusion (page 9)

“Although the fed or semi-fed ticks may be newly infected, since it is not known whether the pathogens’ DNA was in a tick or from the animal's blood, all ticks collected from the region of Pleven, were collected from the vegetation by flagging and were unfed. Respectively, we may accept the Pleven region as a site with a higher prevalence of Anaplasmataceae naturally infected ticks than the Black Sea coast region. In addition,…”

  1. Hence, I would rethink the paper title. In my opinion it should be changed.

Authors: The paper title was changed to:

Anaplasma and Ehrlichia species in Ixodidae ticks collected from two regions of Bulgaria”
